# Surface Defect Mitigation of Additively Manufactured Parts Using Surfactant-Mediated Electroless Nickel Coatings

**DOI:** 10.3390/ma17020406

**Published:** 2024-01-13

**Authors:** Anju Jolly, Véronique Vitry, Golnaz Taghavi Pourian Azar, Thais Tasso Guaraldo, Andrew J. Cobley

**Affiliations:** 1The Functional Materials and Chemistry Group, Centre for Manufacturing and Materials, The Institute for Advanced Manufacturing and Engineering, Coventry University, Beresford Ave., Coventry CV6 5LZ, UK; ac8637@coventry.ac.uk (G.T.P.A.); ae1526@coventry.ac.uk (T.T.G.); aa2266@coventry.ac.uk (A.J.C.); 2Department of Metallurgy, University of Mons (UMONS), 23 Place du Parc, B-7000 Mons, Belgium; veronique.vitry@umons.ac.be

**Keywords:** electroless deposition, nickel, additive manufacturing, levelling, filling, surfactants

## Abstract

The emergence of defects during the early production phases of ferrous-alloy additively manufactured (AM) parts poses a serious threat to their versatility and adversely impacts their overall mechanical performance in industries ranging from aerospace engineering to medicine. Lack of fusion and gas entrapment during the manufacturing stages leads to increased surface roughness and porosities in the finished part. In this study, the efficacy of employing electroless nickel–boron (Ni-B) deposition to fill and level simulated AM defects was evaluated. The approach to levelling was inspired by the electrochemical deposition techniques used to fill vias in the electronics industry that (to some extent) resemble the size and shape of AM-type defects. This work investigated the use of surfactants to attenuate surface roughness in electroless nickel coatings, thereby achieving the preferential inhibition of the coating thickness on the surface and promoting the filling of the simulated defects. A cationic surfactant molecule, CTAB (cetyltrimethyl ammonium bromide), and a nonpolar surfactant, PEG (polyethylene glycol), at different concentrations were tested using a Ni-B electrolyte for the levelling study. It was found that the use of electroless Ni-B to fill simulated defects on ferrous alloys was strongly influenced by the concentration and nature of the surfactant. The highest levelling percentages were obtained for the heavy-molecular-weight PEG-mediated coatings at 1.2 g/L. The results suggest that electroless Ni-B deposition could be a novel and facile approach to filling defects in ferrous-based AM parts.

## 1. Introduction

Additive manufacturing (AM) is the layer-by-layer fabrication of three-dimensional articles governed by computer-aided design (CAD) software data. Owing to its high geometric accuracy, complex spatial profiles, and enhanced performance, AM parts find applications in fields ranging from medicine to aerospace engineering [1]. Contrary to subtractive manufacturing practices, the construction of intricate structures with composite materials using the “bottom-up” approach is revolutionising the manufacturing industry by accelerating production and increasing cost-effectiveness [2,3]. Among different techniques, metal AM parts are widely fabricated using laser powder bed fusion (LPBF) processes. However, the versatility of AM is held back by the recurrent defect formations in its early production stages. In the case of LPBF AM parts, defects can include unintended rough surfaces, increased porosity, and design deviations. While the overall bulk porosity can be controlled, individual pore size and shape can cause failure in the fabricated part. It has been suggested that a bulk porosity of less than 1% can have a limited impact on the mechanical properties of AM parts [4]. However, a reduction in porosity shifts the reliance of mechanical properties to critical factors such as surface roughness, microstructure, and AM geometry. These defects can lead to reduced ductility and lower tensile strength, while an increase in pore size can cause a reduction in elongation and decreased yield strength. In addition, pores near the surface impact fatigue properties [5]. The synergistic effects of these defects alter the overall mechanical performance of the AM parts.

In the AM industry, techniques to eliminate defects in the manufacturing stages have been tested, and these remain an active area of research today. To minimise pore defects, Yadroitsev et al. tried to optimise processing parameters and found that the optimum hatch space—which is the distance between two consecutive laser beams—of 120 µm, while taking the laser width to be 70 µm, minimizes porosity [5]. In another study by Kobryn et al., the lack of fusion and gas porosities decreased with an increasing scan speed and power level [6]. Another method was proposed by Li [7], where the feedstock baking immediately before the AM fabrication achieved positive results in spherical porosity reduction. However, optimising the process parameters to remove defects is a very time-consuming process, and while it may substantially reduce of fusion defects, the problem of gas porosities persists. To summarise, these studies were able to identify that gas porosities were caused by issues with feedstock material as well as powder delivery processes, while the lack of fusion defects was dependent on process parameters [8]. In addition, post-treatment methods [9] such as hot isostatic pressing (HIP), a technique that involves the simultaneous application of both high temperature and pressure (isostatic) to collapse and diffusion-bond internal pores in a structure, have been employed for densification and bulk porosity reduction. However, this approach suffers from lowering the ductility of the AM parts as well as damaging their stress rupture response [10,11]. Properties such as ductility and stress response are important in the automotive and aerospace industries, in conjunction with a requirement for substantial anticorrosive behaviour and fatigue resistance [1,12,13]. In a study by Masuo et al., it was reported that while HIP could eliminate a significant amounts of the AM defects while improving fatigue performance, the surface roughness of the finished parts could not be reduced. Additionally, in recent years, laser shock peening has been employed to investigate the near-surface pore (surface defect) closure effect in AM components [14]. However, this process is limited by the laser shock peening power because it is not viable to employ higher powers, as it can damage the surface of the AM parts. This technique also suffers from high equipment costs and attaining uniform coverage for complex shapes as it is a line-of-sight process. Hence, the issue of defects remains a challenge in the AM industry.

The metal AM industry is predicted to have an exponential growth of around USD 11.45 billion by the year 2030 [15]; therefore, research into the mitigation of these defects is of significant academic and industrial interest. One such approach is electrochemical deposition. As a surface finishing technique, electrochemical deposition finds applications in sectors such as electronics, aerospace, and automotive. Of particular interest is the filling of blind vias and ultra-large-scale integration (ULSI), a process used in the production of integrated circuits. ULSIs and vias have similar geometries to the defects and voids found in AM parts, and so the electrochemical deposition techniques utilised to fill them are of particular interest in this study. This study has focused on the use of Nickel for defect filling of ferrous alloy components and the employment of electroless Nickel coatings on these materials has the advantage that pretreatments such as the Wood’s nickel strike required when electrodeposition is utilised, is not necessary. For electroless deposition, the “filling” action is often acquired by the addition of surfactants or additives. There have been many advances in the literature on electroless copper deposition to fill ULSI [16,17,18,19]; however, electroless nickel deposition has garnered fewer studies. One such study is where L. Zan et al. reported a sub-micrometre filling using an electroless nickel phosphorous (Ni-P) coating with heavy-molecular-weight surfactant molecules [20] on a SiO_2_ surface. Nickel coatings, especially those of electroless nickel–boron (Ni-B), can alter the surface morphologies of metal substrates and even enhance surface characteristics such as corrosion resistance, fatigue properties, etc. [21,22]. Moreover, they have been used in various industrial sectors for the past 40 years owing to their excellent tribological and mechanical properties, which are far superior to those of hard chrome coatings [23]. In addition, it was postulated that the use of a nickel coating on a ferrous-alloy substrate would not significantly alter the composition of the finished part whilst producing a smoother, defect- and void-free material. Therefore, the present study introduces a first-hand investigation into the levelling behaviour of surfactant-mediated Ni-B coatings on AM-type defects.

## 2. Materials and Methods

### 2.1. Substrate Preparation

For substrates, mild steel ST 37-DIN 17100 with dimensions of 5.0 × 2.5 × 1.0 cm^3^ was used. The substrates were prepared by polishing with SiC papers of 240 and 800 grit. Prior to activation, they were cleaned in acetone for 1 min before being activated by immersing in 30 vol.% HCl acid for 1 min. The substrates were rinsed after each step with deionised water. They were then immediately placed in the electrolyte for 1 h of deposition after activation.

### 2.2. Electroless Nickel

The CTAB-mediated coatings were prepared by varying the concentration of CTABfrom 0.3 to 1.2 g/L with 0.3 g/L increments of surfactant using SnCl_2_ as a stabiliser (adapted from the work of Bonin et al. [24]) as shown in Table 1. For PEG-mediated coatings, only the lower SnCl_2_ addition results are in the present study since the higher stabiliser concentrations resulted in complete inhibition of nickel deposition.

### 2.3. Simulation of AM Defects

In order to test the feasibility of various electroless deposition compositions, to fill and level AM defects, a baseline defect morphology with reproducible depth had to be established. For this purpose, real AM parts could not be utilised because the defect position, depth, shape, and overall geometry would not be consistent or predictable. This led to the development of a methodology to simulate AM-type defects on ferrous-alloy substrates. The simulation of AM-type defects on various ferrous-based substrates was achieved by using a microhardness tester (using micro loads). By varying the indenter profile, a Vickers four-sided pyramid diamond indenter, and a Knoop pyramidal diamond point, and by keeping the load constant, defects of varying depth and shape were obtained. The Table 2 represents the two test conditions applied to produce reproducible defects.

All the electroless experiments were carried out on samples with 6 simulated defects, where 3 were of Knoop profile and the rest were Vickers. A uniform load of 1 Kgf (98.1 N) was applied to create all the simulated defects.

### 2.4. Levelling Analysis Using 3D Optical Profile Measurements

The levelling percentage was determined using Equation (1) and the 3D optical microscope depth data collected pre and post deposition. The additive inverse of the levelling percentages were plotted to determine the negative and positive levelling.
(1)$$\mathrm{Levelling}\;\mathrm{Percentage}=\frac{\mathrm{Final}\;\mathrm{depth}\;-\;\mathrm{Initial}\;\mathrm{depth}}{\mathrm{Initial}\;\mathrm{depth}}\times 100$$

### 2.5. Characterisation

A HIROX KH-8700 (Tokyo, Japan) digital optical microscope was used to determine the coating growth and levelling behaviour. The plating thickness and nucleation of coating over the simulated defects were measured by setting the sectioned samples in a resin and polishing them up to a mirror finish using a modified metallographic procedure.

A Zeiss 119 Surfcom 1400D-3DF (Oberkochen, Germany) apparatus was employed to measure the deposit surface roughness, namely, average surface roughness value (Ra), and the reported value is the mean of 3 samples, with 5 measurements recorded per sample. The average initial surface roughness of the mild steel substrates with defects is 0.34 ± 0.14.

A Hitachi SU8020 (Tokyo, Japan) scanning electron microscope (SEM) was used to analyse the surface and the cross-section morphology of each sample.

The simulated defects were prepared using a Mitutoyo HM-200 (Kawasaki, Japan) microhardness tester.

## 3. Results and Discussion

### 3.1. Levelling Analysis

Figure 1a shows the depth profile of the defect after electroless deposition without surfactant addition and depicts a uniform increase in apparent defect depth for the Knoop defect geometry (similar behaviour was observed for Vickers), leading to a negative levelling percentage, as shown in Figure 2 and Figure 3. At 0.6 g/L of CTAB addition, the defect depth for sample (c) showed the highest positive levelling percentage for the Knoop defects. The 3D optical micrographs in Figure 1a–e show a deviation from the typical agglomerated cauliflower structures seen with electroless Ni-B coatings over the top of the defects, especially for 0.3 and 0.6 g/L additions of CTAB. A surfactant-influenced variation in the morphology as well as defect depths is evident from the micrographs. Surface roughness attenuation of NiP deposits using CTAB has been reported [25] where, with smaller increments of CTAB concentration, a decline in average surface roughness was observed. The current study also follows this trend of decreasing surface roughness, and the average surface roughness is discussed in detail in Section 3.3. Additionally, it can be inferred that addition of 0.3 g/L and 0.6 g/L CTAB aided modification in the morphology and marginally reduces the defect depth for Knoop and Vickers defect geometries, respectively.

Further study on the influence of lower surface roughness on levelling was conducted. Based on a roughness attenuation study for Ni-B [24], a higher concentration of SnCl_2_ was used to study its impact on levelling. From Figure 1f–j, it is clear that the increase in stabiliser concentration modified the surface; however, the CTAB-aided roughness attenuation did not positively affect levelling, as shown by the levelling percentages of both Knoop and Vickers defects in Figure 2 and Figure 3, respectively.

In order to study the impact of surface modification and levelling effect of heavy-molecular-weight surfactant molecules [20] in Ni-B deposition, varying concentrations of PEG were tested on simulated defects. In Figure 1k–o, the optical micrographs for the defects after deposition are shown for varying concentrations of PEG with 0.06 g/L of SnCl_2_. A concentration-dependent surface modification effect in the coatings and the depth of the simulated defect is observed. As for the levelling effect, in the case of Knoop defect morphology, it can be seen that it is the 1.2 g/L addition that resulted in the best fill, according to Figure 2. However, for Vickers defect geometry, the highest percentage of levelling was observed for the 0.6 g/L addition of PEG (shown in Figure 3). In addition, levelling percentages for different concentrations are within the error range.

To summarise the levelling study, the surfactant-enabled levelling of Ni-B using higher-molecular-weight surfactant PEG was able to achieve a much better fill than the lower-molecular-weight CTAB surfactant. An attempt was made to study the relation between CTAB-aided coatings for roughness attenuation and their impact on levelling. However, although the stabiliser SnCl_2_ helped with lowering the surface roughness, it did reduce the defect depth.

### 3.2. Surface Morphology and Microstructure

The surface morphology results are reported using Figure 4.

From the surface morphology study, a noticeable change in the surface roughness, detailed in Section 3.3 below, is observed in the SEM images. For a 0.06 g/L concentration of stabiliser, there is a visible deviation and loss of compactness in the cauliflower structure formation of the coatings. Smaller clusters, for a 0.06 g/L addition of stabiliser, are observed in the 0.9 g/L PEG samples. Furthermore, these samples appear low in surface roughness as well. However, with the increased addition of CTAB, at 1.2 g/L, a highly agglomerated coating morphology is observed. Similarly, for the 0.2 g/L stabiliser samples, more compact cauliflower structures are observed. These coatings have a low surface roughness [24] compared to their counterpart (0.06 g/L samples). However, at higher concentrations of CTAB, an agglomerated rough coating was produced, similarly to the 0.06 g/L SnCl_2_ samples. This sudden change in the surface morphology could be due to the formation of micelles. Vijayanand et al. [26] reported that the micelle formations can lead to a reduction on the wettability and coalescence of nickel particles. In contrast, in the case of PEG, the results show that with increasing concentrations, the coating appears to deviate from the conventional cauliflower structure toward a uniform coating, with the lowest surface roughness reported for the 1.2 g/L addition. It was reported by S. Afroukhteh et al. [27] that the incorporation of PEG at a concentration of 1 g/L was able to produce coatings with a smoother morphology in the case of electroless nickel–phosphorous coatings. A similar behaviour was observed in the case of Ni-B.

### 3.3. Roughness

In the case of electroless coatings, substrate surface roughness is a key factor that influences the evolution of the final roughness of the coatings. This can be a crucial factor to consider while trying to mitigate surface roughness in AM parts, as AM-built parts tend to have a surface roughness in the order of 5–40 µm depending on the type of AM process [28,29]. Due to an increase in stabiliser concentration, as reported in [24], the average surface roughness (Ra) value was brought down to <1 µm (with the exception of the 1.2 g/L CTAB sample), as shown in Figure 5. However, no positive levelling was observed for this set of samples.

In addition, this study recognised heavy-molecular-weight PEG as an optimal surfactant for reducing surface roughness and achieving positive levelling compared to a surfactant-free NiB coating. Notably, the addition of PEG at 1.2 g/L yielded the lowest Ra value and the highest positive levelling. It is crucial to highlight that, owing to the electroless deposition method utilised in this study, the surface roughness of the coating typically mimics the profile of the substrate but consistently exhibits a higher value of surface roughness.

### 3.4. Microsection and Levelling Behaviour

Conventional electroless deposition would produce a uniform coating over the substrate surface, especially over complex shapes. However, the optical micrographs as well as the cross section of defects revealed that the evolution of coating thickness over the defect depths and the defect edges (top of the defects) differs. Figure 6a–c, illustrate the expected and real cross-sectional coating morphology over both Vickers and Knoop defects for a 0.06 g/L addition of stabiliser. For a uniform load, the morphology of the defect surface produced varies according to the indent profiles. In addition to depth data detailed in Section 3.1, the width of Knoop to Vickers varies from 75 µm to 135 µm. Similarly, the angle of the defect walls varies from 128° to 148°, respectively. As shown in Figure 6a, due to the unique cauliflower-tree-like morphology of the NiB coating with a tin stabiliser [24], the coating over the defects was expected to evolve and fill the defects with a uniform-thickness coating, and the Knoop indent defects were expected to have a better fill. However, the real cross-section of the indent defects, as shown in Figure 6b,c, illustrates that the evolution of the coating thickness over the bottom of the defects and the top edges varies significantly. In the bottom of the defect, a smoother morphology with a thinner coating is observed, whilst the coating at the top edges of the defect shows more agglomerated cauliflower structures of a higher thickness. Furthermore, negative levelling was reported for the higher-stabiliser-concentration samples with 0.2 g/L of various CTAB additions, as seen in Figure 2 and Figure 3. This finding was further confirmed through the cross-sectional SEM images shown in Figure 6d,e. Here, although the surface roughness was significantly reduced and led to the formation of compact cauliflower structures, the growth of the coating thickness over the bottom of the defect appears inhibited. It is imperative to note that, as shown in Figure 7, although an increase in CTAB mediation at a higher stabiliser concentration increased the rate of deposition on the surface, it did not accelerate the growth of the coating thickness at the bottom of the defects, resulting in overall negative levelling.

Finally, with the PEG additions, the cross-sections offer a much more distinct coating nucleation compared with their CTAB counterparts. In conjunction with attenuating the surface roughness of the coating, PEG at 1.2 g/L was more able to enhance the deposition rate at the bottom of the defect compared to the surface. As shown through the percentage levelling study, the highest levelling was observed for the PEG-aided coating for both the Vickers and the Knoop defects. It was reported by L. Zan et al. that a heavy-molecular-weight surfactant has a low diffusion rate and, therefore, it is more likely to adsorb at the surface of the substrate than the bottom of the defect. This creates a concentration gradient. While the polymeric surfactant inhibits deposition at the top as well as the bottom of the defect, the rate of this inhibition effect varies due to the concentration gradient absorption of the compound on the uneven (defect) surface, as shown in Figure 8. Therefore, this leads to a higher deposition rate at the bottom compared to the top of the defect which, in turn, leads to filling and levelling [20]. This concentration-dependent behaviour of PEG, the heavy-molecular-weight surfactant, is evidenced in the percentage levelling studies reported in Figure 2 and Figure 3. Moreover, Figure 8 depicts the PEG levelling mechanism, as shown in the cross-sectional SEM images in Figure 6f,g, on simulated defects and its potential to facilitate positive levelling on real AM defects, specifically keyhole-type defects [30], owing to the depth-to-width ratio being much larger than the simulated one.

## 4. Conclusions

The evolution of coating thickness and morphology over the simulated defect depths as well as the top portion of the defect is non-uniform. This has a significant impact on the levelling and filling of these simulated defects.

The surfactant CTAB can bring about some levelling and filling in Ni-B coatings (especially with 0.9 and 0.6 g/L additions). However, it was the PEG that was able to demonstrate the best levelling for both the Vickers and Knoop geometries.

Surface roughness and coating morphology can be modified using both the surfactants. The lowest surface roughness was reported for the 1.2 g/L addition of PEG (<1 µm).

For coatings deposited from an electrolyte with a high stabiliser concentration, a uniform increase in apparent defect depth was observed after electroless deposition. An inverse relationship between surface roughness and levelling was observed with this stabiliser concentration.

The heavy-molecular-weight surfactant PEG exhibited a concentration-dependent levelling behaviour, with the higher concentrations exhibiting the highest percentage of defect depth change.

In comparison, PEG at 1.2 g/L addition can be a good candidate for levelling in electroless NiB.

In future work, alternate heavy-molecular-weight nonpolar surfactants will be incorporated into the electrolyte to test the preferential growth, inhibition, and levelling of simulated defects.

## Figures and Tables

**Figure 1 materials-17-00406-f001:**
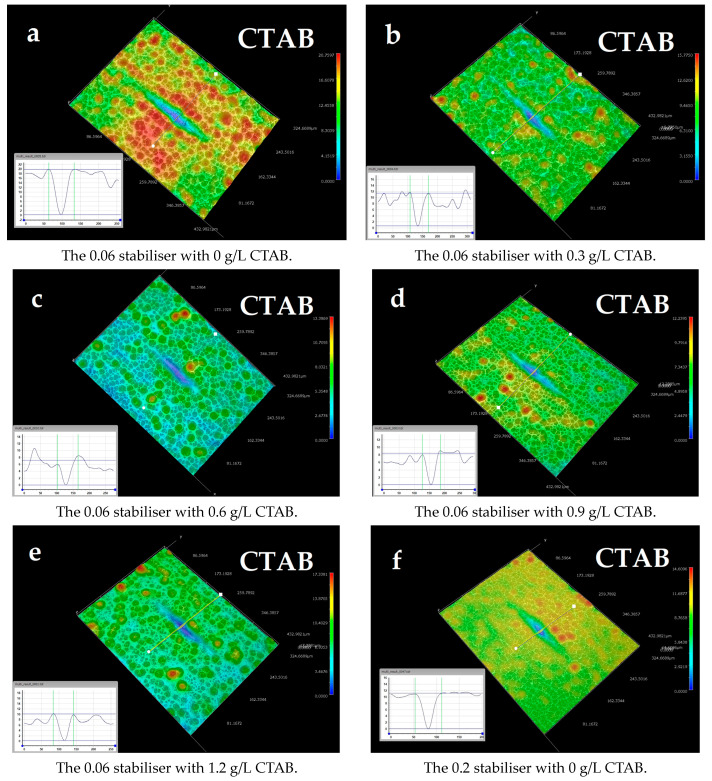
Three-dimensional optical micrographs of samples depicting the variation in morphology of deposits for Knoop indent defects with varying surfactant and stabiliser concentrations for CTAB and PEG.

**Figure 2 materials-17-00406-f002:**
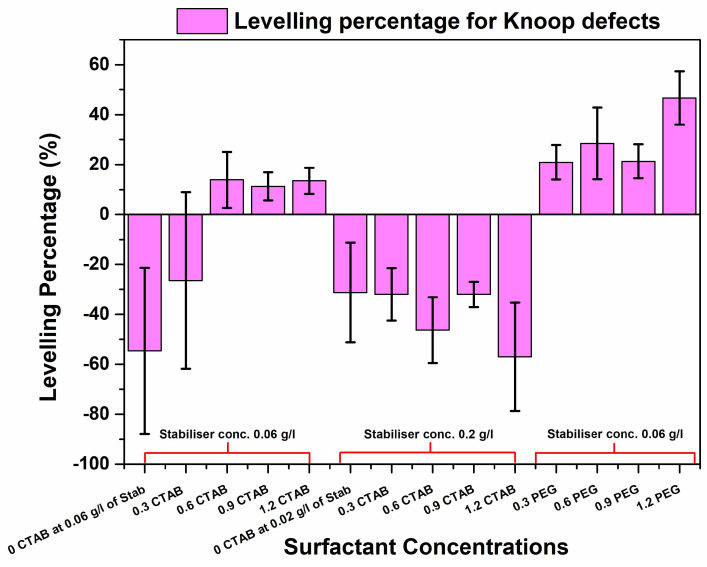
Levelling percentages for Knoop defects depicting the positive and negative levelling achieved using PEG and CTAB with two different SnCl_2_ (stabiliser) concentrations.

**Figure 3 materials-17-00406-f003:**
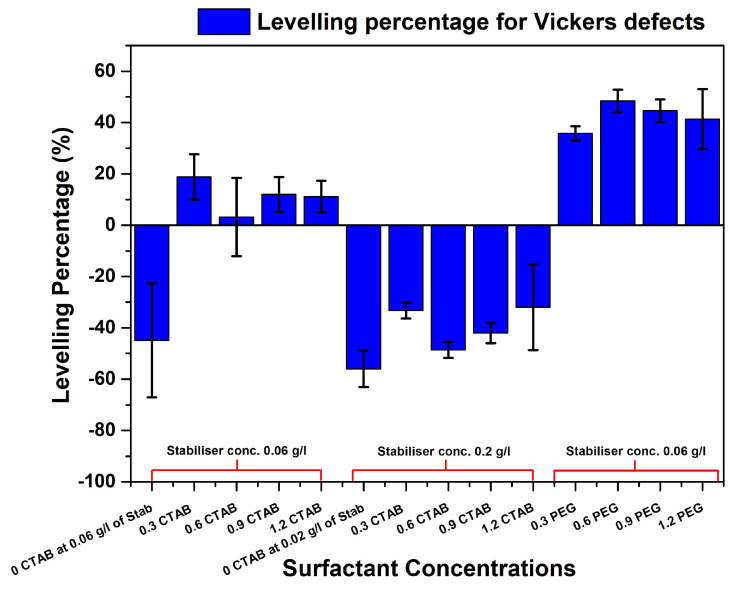
Levelling percentages for Vickers defects depicting the positive and negative levelling achieved using PEG and CTAB with two different SnCl_2_ (stabiliser) concentrations.

**Figure 4 materials-17-00406-f004:**
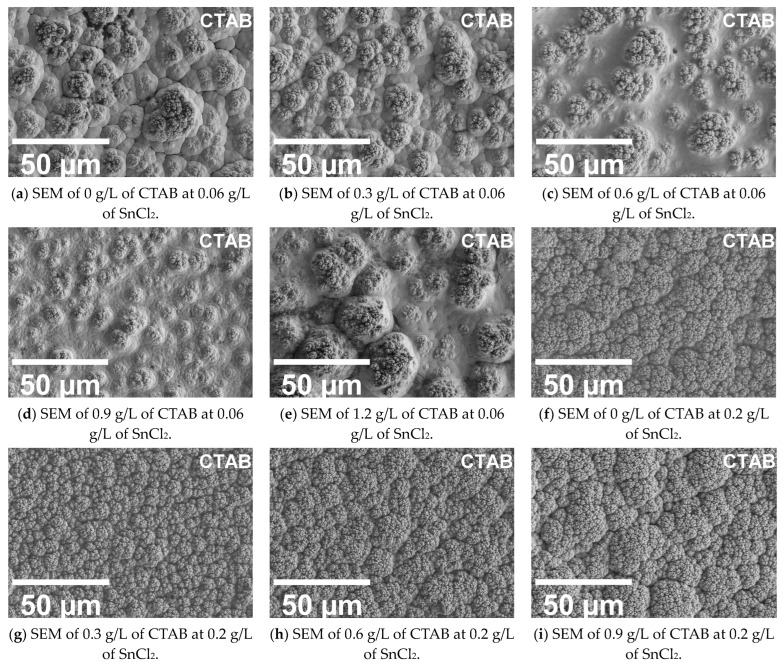
SEM images of electroless coatings for varying surfactant and stabiliser concentrations for CTAB and PEG.

**Figure 5 materials-17-00406-f005:**
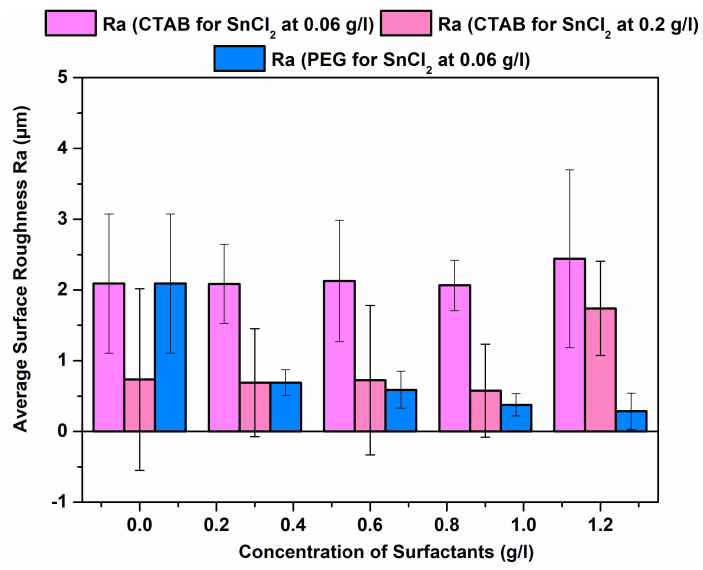
Comparison of Ra values for coatings with various surfactant additions (PEG and CTAB) for low (0.06 g/L) and high (0.2 g/L) stabiliser concentrations.

**Figure 6 materials-17-00406-f006:**
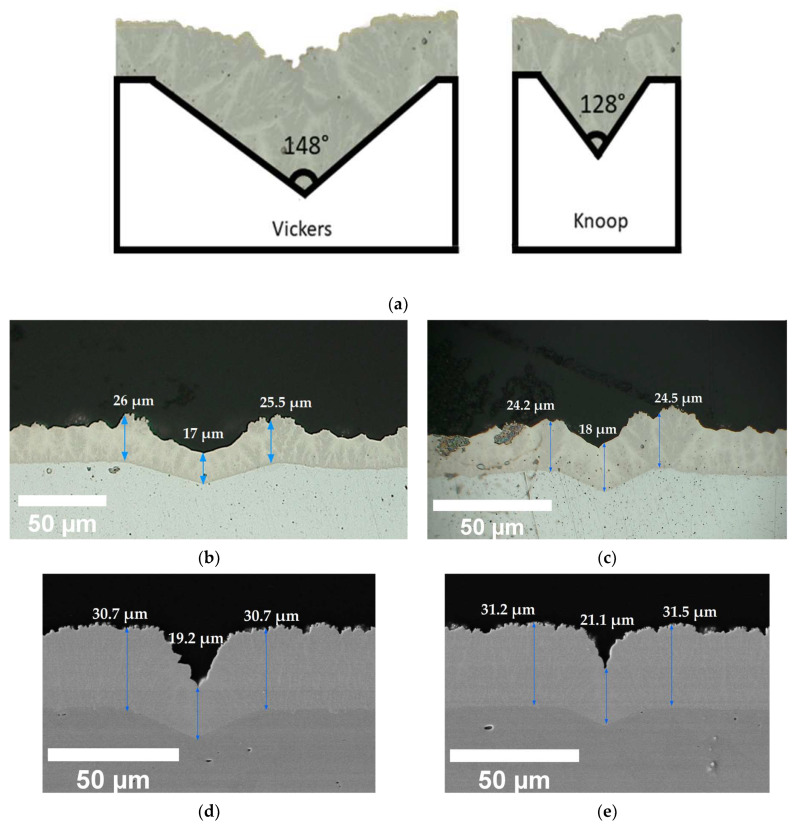
Diagrammatic representation (**a**) of expected nucleation in NiB electroless coating over the cross-sections of Vickers and Knoop indents. Polished microsections (optical images) of (**b**) Vickers and (**c**) Knoop indent defects for 0.3 g/L CTAB concentration with 0.06 g/L stabiliser concentration and cross-sectional SEM images of (**d**) Vickers and (**e**) Knoop indent defects for 1.2 g/L CTAB concentration with 0.20 g/L stabiliser concentration. PEG additions of 1.2 g/L (**f**) Vickers and (**g**) Knoop indent defects for with 0.06 g/L stabiliser concentration.

**Figure 7 materials-17-00406-f007:**
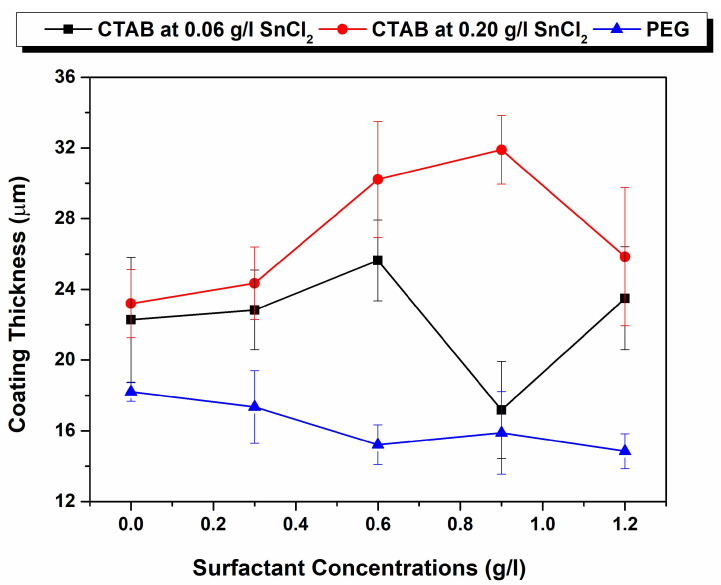
Comparison of coating thicknesses measured using cross-section of sample plated with stabiliser concentrations 0.06 g/L and 0.2 g/L of CTAB along with PEG with stabiliser concentration of 0.06 g/L.

**Figure 8 materials-17-00406-f008:**
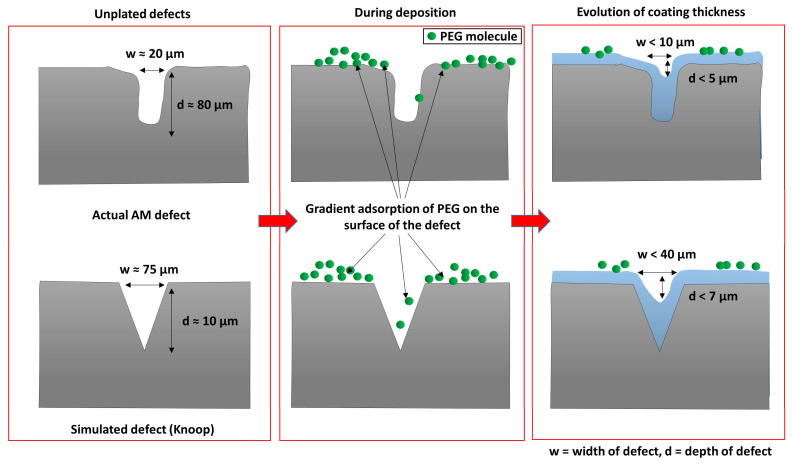
PEG-aided levelling mechanism on simulated and AM defects.

**Table 1 materials-17-00406-t001:** Composition of tin-stabilised electroless NIB.

Compound	
NiCl_2_·6H_2_O (g/L) (99%—VWR Chemicals, Radnor, PA, USA)	24
NaBH_4_ (g/L) (99.9%—Acros Organics, Fair Lawn, NJ, USA)	0.602
NH_2_-CH_2_-CH_2_-NH_2_ (mL) (99%—VWR Chemicals, Radnor, PA, USA)	59
NaOH (g/L) (VWR Chemicals, Radnor, PA, USA)	39
SnCl_2_ (g/L) (99.9%—Acros Organics, Fair Lawn, NJ, USA)	Variable
CTAB (g/L) (99%—VWR Chemicals, Radnor, PA, USA)	Variable
PEG (g/L) (Mw~6000—VWR Chemicals, Radnor, PA, USA)	Variable

**Table 2 materials-17-00406-t002:** Comparison of simulated defect depth and applied force on mild steel substrates (HK represents Knoop and HV represents Vickers).

Test Condition	Applied Force (N)	Average Depth (Mild Steel Substrate) (μm)
1HK	9.81	10.40 ± 1.20
1HV	9.81	15.45 ± 0.89

## Data Availability

The data presented in this study will be made available upon request from the corresponding author.

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
