# Peer review of "Surface Defect Mitigation of Additively Manufactured Parts Using Surfactant-Mediated Electroless Nickel Coatings"

_materials, 2024, doi:10.3390/ma17020406_

Round 1
Reviewer 1 Report
Comments and Suggestions for Authors
The theme of the manuscript is very interesting especially on the defect filling in additively manufactured parts using electroless nickel boron coatings. While the trials on simulated conditions is understandable, the filling of defects on those simulated parts with a very well known electroless nickel boron chemistry is not significant. Poly ethylene glycol and CTAB are well known additives that act as levellers. What differs from previous studies needs to be highlighted in the manuscript and performance data needs to be presented.I would recommend for a thorough revision with the motivation , performance data, schematic showing the performance before and after defect filling etc,
Reviewer 2 Report
Comments and Suggestions for Authors
1-The title of this paper does not truly reflect the essence of this work. No additively manufactured parts were used and instead defects were imparted using indentation.
2-Using indentation only does not truly reflect the morphology of the defects you see after additive manufacturing. How do you explain this?
3- What was the surface roughness and how do you compare this surface roughness with actual additively manufactured component?
4- Line 224: Author claims that surface roughness in additive manufacturing ranges from 10-40microns. This is untrue. With a slight optimization, the surface roughness of 3microns or even less can be accomplished and it is based on my personal experience and previous works in additive manufacturing.
5- How is the strength of bonding between coating and substrate? Did authors do any mechanical testings such as nanoindentation, or wear/scratch testing to examine it?
6-How is the corrosion resistance of the coating layer?
7- From practicality point of view, the surface of additively manufactured parts are covered with a thin oxide layer. Would it be feasible to coat these parts with your eleteroless process without surface preparation?
Comments on the Quality of English Language
1-The title of this paper does not truly reflect the essence of this work. No additively manufactured parts were used and instead defects were imparted using indentation.
2-Using indentation only does not truly reflect the morphology of the defects you see after additive manufacturing. How do you explain this?
3- What was the surface roughness and how do you compare this surface roughness with actual additively manufactured component?
4- Line 224: Author claims that surface roughness in additive manufacturing ranges from 10-40microns. This is untrue. With a slight optimization, the surface roughness of 3microns or even less can be accomplished and it is based on my personal experience and previous works in additive manufacturing.
5- How is the strength of bonding between coating and substrate? Did authors do any mechanical testings such as nanoindentation, or wear/scratch testing to examine it?
6-How is the corrosion resistance of the coating layer?
7- From practicality point of view, the surface of additively manufactured parts are covered with a thin oxide layer. Would it be feasible to coat these parts with your eleteroless process without surface preparation?
Reviewer 3 Report
Comments and Suggestions for Authors
The paper is generally well-written and interesting, but the following issues should be addressed:
1) what is the surface roughness of the specimen without coating (with both Knoop and Vickers dent)? This is necessary to understand the effect of the coating. Add the surface profile to Figure 1.
2) You presented the roughness value range: How did you get that range? Did you test multiple samples or did you take multiple readings on one sample? This is not clear in your research methodology.
3) Table 1 is very confusing, not sure what is in the second column. Where are the first 4 chemicals used?
4) It is obvious that different combinations of chemicals will lead to different thickness and texture of the coating. However, the thickness is almost constant across the whole surface (see Figure 6), with some variation in surface texture as shown in Figure 4. In my opinion, leveling percentages would be very meaningless without addressing what I mentioned. It would be more interesting to quantify the thickness of the coating (not leveling percentages) due to variations in chemical composition. This can be achieved from SEM photos (a lot of which is not included).
Round 2
Reviewer 1 Report
Comments and Suggestions for Authors
The authors have now highlighted the motivation behind this work and can be accepted for publication
Reviewer 3 Report
Comments and Suggestions for Authors
More samples are needed, the authors have only done one sample per condition, which may not be representative of their work.
